# Cluster & Tune: Enhance BERT Performance in Topical Text Classification when Labeled Data is Scarce

## Abstract

In cases where labeled data is scarce, the common practice of fine-tuning BERT for a target text classification task is prone to producing poor performance. In such scenarios, we suggest performing an unsupervised classification task prior to fine-tuning on the target task. Specifically, as such an intermediate task, we perform unsupervised clustering, training BERT on predicting the cluster labels. We test this hypothesis on various data sets, and show that this additional classification step can significantly reduce the demand for labeled examples mainly for topical classification tasks. We further discuss under which conditions this task is helpful and why.

## 1 Introduction

One of the most practical NLP use cases is the task of text classification, where the goal is to automatically assign a new text instance into a subset of pre-specified categories. Text classification applications include topic detection, sentiment analysis, and spam filtering, to name just a few examples. The standard paradigm relies on supervised learning, where it is well known that the size and quality of the labeled data strongly impact the performance of the trained classifier. Hence, as with many other supervised learning tasks, developing a text classification scheme in practice, typically requires to make the most out of a relatively small set of annotated examples.

The emergence of transformer-based pretrained language models such as BERT (Devlin et al., 2018) has reshaped the NLP landscape, leading to significant advances in the performance of most NLP tasks, text classification included (e.g., Nogueira & Cho, 2019; Ein-Dor et al., 2020). These models typically rely on pretraining a transformer-based neural network on massive and heterogeneous corpora on a general Masked Language Modeling (*MLM*) task, i.e., predicting a word that is masked in the original text. Later on, the obtained model is fine-tuned to the actual task of interest, termed here the *target task*, using the labeled data available for this task. Thus, pretrained models serve as general sentence encoders which can be adapted to a variety of tasks (Lacroix et al., 2019; Wang et al., 2020a).

Our focus in this work is on a challenging yet common scenario, where the textual categories are not trivially separated; furthermore, the available labeled data is scarce. There are many real-world cases in which data cannot be sent for massive labeling by the crowd (e.g., due to confidentiality of the data, or the need for a very specific expertise) and the availability of experts is very limited. In such setting, fine-tuning a pretrained model is expected to yield far from optimal performance. To overcome this, one may take a gradual approach composed of various steps. One possibility is to further pretrain the model with the *self-supervised* MLM task over unlabeled data taken from the target task domain (Whang et al., 2019). Alternatively, one can train the pretrained model using a *supervised* intermediate task which is different in nature from the target-task, and for which labeled data is more readily available (Pruksachatkun et al., 2020; Wang et al., 2019a; Phang et al., 2018). Each of these steps is expected to provide a better starting point – in terms of the model parameters – for the final fine-tuning step, performed over the scarce labeled data available for the target task, aiming to end up with improved performance.

Following these lines, here we propose a simple strategy, that exploits *unsupervised* text clustering as the intermediate task towards fine-tuning a pretrained language model for text classification. Our

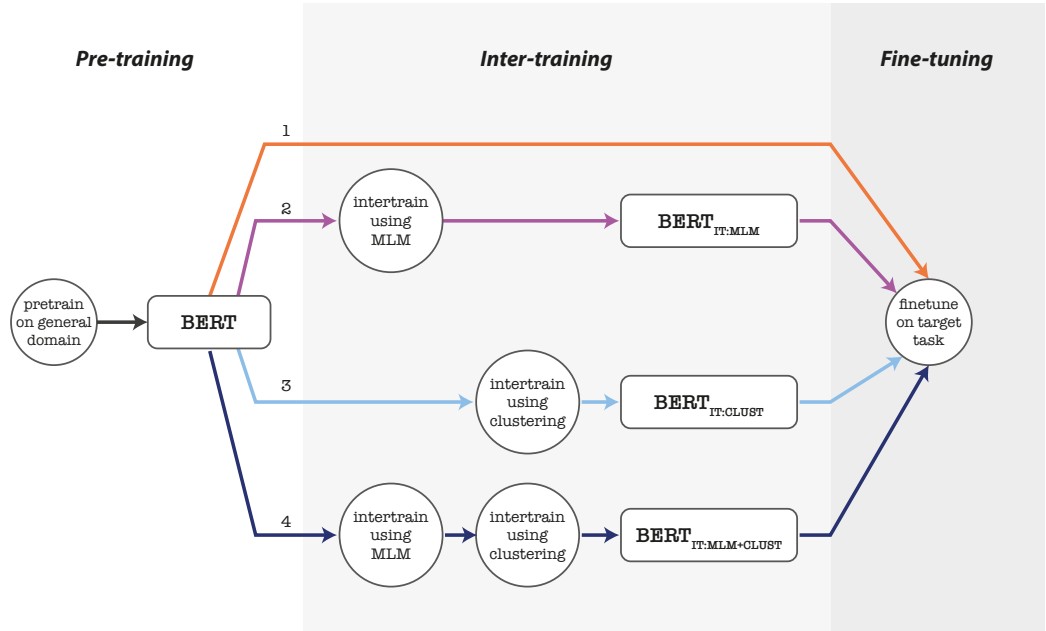

Figure 1: BERT phases - circles are training steps which produce models, represented as rectangles. In the pre-training phase, only general corpora is available. The inter-training phase is exposed to target domain data, but not to its labeled instances. Those are only available at the fine-tuning phase.

work is inspired by the use of clustering to obtain labels for training deep networks in computer vision (Gidaris et al., 2018; Kolesnikov et al., 2019). Specifically, we use an efficient clustering technique, that relies on simple Bag Of Words (BOW) representations, to partition the unlabeled training data into relatively homogeneous clusters of text instances. Next, we treat these clusters as labeled data for an intermediate text classification task, and train BERT – with or without additional MLM pretraining – with respect to this multi-class problem, prior to the final fine-tuning over the actual target-task labeled data. Extensive experimental results demonstrate the practical value of this strategy on a variety of benchmark data, most prominently when the training data available for the target task is relatively small and the classification task is of topical nature. We further analyze the results to gain insights as to when this approach would be most valuable, and propose future directions to expand the present work.

## 2 INTERMEDIATE TRAINING USING UNSUPERVISED CLUSTERING

A pre-trained transfer model, such as BERT, is typically developed in consecutive phases. First, the model is *pretrained* over massive general corpora with the MLM task.[1] The obtained model is referred to henceforth as *BERT*. Second, BERT is *finetuned* in a supervised manner with the available labeled examples for the target task at hand. This standard flow is represented via Path-1 in Fig. 1.

An additional phase can be added between these two, referred to next as *intermediate training*, or inter-training in short. In this phase, the model is exposed to the corpus of the target task, or a corpus of the same domain, but still has no access to labeled examples for this task.

A common example for such an intermediate phase is to continue to intertrain BERT using the self-supervised MLM task over the corpus or the domain of interest, sometimes referred to as further or adaptive pre-training (e.g., Gururangan et al., 2020). This flow is represented via Path-2 in Fig. 1, and the resulting model is denoted $BERT_{IT:MLM}$, standing for Intermediate Task: MLM.

---

[1]BERT was originally also pretrained over a "next sentence prediction" task (Devlin et al., 2018); however, later works such as Yang et al. (2019) and Liu et al. (2019b) have questioned the contribution of this additional task and focused on MLM.

|  | Train | Dev | Test | # classes | Topical |
|---|---|---|---|---|---|
| Yahoo! answers | 15,000 | 3,000 | 3,000 | 10 | Yes |
| DBpedia | 15,000 | 3,000 | 3,000 | 14 | Yes |
| CFPB | 15,000 | 3,000 | 3,000 | 15 | Yes |
| 20 newsgroups | 10,182 | 1,132 | 7,532 | 20 | Yes |
| AG's news | 15,000 | 3,000 | 3,000 | 4 | Yes |
| ISEAR | 5,366 | 766 | 1,534 | 7 | Yes |
| SMS spam | 3,900 | 557 | 1,115 | 2 | No |
| Subjectivity | 7,000 | 1,000 | 2,000 | 2 | No |
| Polarity | 7,463 | 1,066 | 2,133 | 2 | No |
| CoLa | 7,592 | 959 | 1,043 | 2 | No |

Table 1: Datasets details: description, number of classes, the split to train/dev/test sets and whether the dataset is topical or not (see details in text).

A key motivation of this paper is to propose a new type of intermediate task, which is designed to be aligned with a text classification target task, and is straightforward to use in practice. The underlying intuition is that inter-training the model over a related text classification task, would be more beneficial compared to MLM inter-training, which focuses on different textual entities, namely predicting the identity of a single token.

Specifically, we suggest *unsupervised* clustering for generating pseudo-labels for inter-training. These pseudo-labels can be viewed as weak labels, but importantly they are not tailored nor require a specific design per target task. Instead, we suggest generating pseudo-labels in a way independent of the target task. The respective flow is represented via Path-3 in Fig. 1. In this flow, we first use unsupervised clustering to partition the training data into $n_c$ clusters. Next, we use the obtained partition as 'labeled' data in a text classification task, where the classes are defined via the $n_c$ clusters, and intertrain BERT over these data to predict the cluster label. In line with MLM, inter-training includes a classifier layer on top of BERT, which is discarded before the fine-tuning stage. The resulting inter-trained model is denoted $BERT_{IT:CLUST}$.

Finally, Path-4 in Fig. 1 represents a sequential composition of Path-2 and Path-3. Thus, in this flow, we first intertrain BERT with the MLM task. Next, the obtained model is further intertrained to predict the $n_c$ clusters, as in Path-3. The model resulting from this hybrid approach is denoted $BERT_{IT:MLM+CLUST}$.

Importantly, following Path-3 or Path-4 requires no additional labeled data, and involves an *a-priori* clustering of training instances that naturally gives rise to an alternative – or an additional – inter-training task. As we show in the following sections, in spite of its simplicity, this strategy provides a significant boost in classification performance in the common scenario, where labeled data for the final fine-tuning is in short supply.

## 3 EXPERIMENTS

### 3.1 TASKS AND DATASETS

We consider 10 datasets that cover a variety of classification tasks and domains. Of these, 6 are topical datasets (including ISEAR which is somewhat borderline), and 4 are non-topical datasets. A topical dataset splits sentences by a high level distinction related to what the sentence is about (e.g., sports vs. economics). On the other hand, non-topical datasets look for finer stylistic distinctions that may depend on the way the sentence is written or on fine details rather than on the central meaning it discusses. It may also separate almost identical sentences, for example, "no" could distinguish between sentences with negative and positive sentiment.

The datasets are: Yahoo! Answers (Zhang et al., 2015), which separates answers and questions to types; DBpedia (Zhang et al., 2015) which differentiates entity types by the beginning of their Wikipedia articles; CFPB[2], which classifies consumer complaints into types; 20 newsgroups (Lang,

---

[2]https://www.consumerfinance.gov/data-research/consumer-complaints/

1995), which includes short documents from 20 Usenet discussion groups; AG's News (Zhang et al., 2015) which contains a classification of news articles; ISEAR (Shao et al., 2015), which considers personal reports for emotion; SMS spam (Almeida et al., 2011), which identifies spam messages; Polarity (Pang & Lee, 2005), which includes sentiment analysis on movie reviews; Subjectivity (Pang & Lee, 2004), which categorizes movie snippets as subjective or objective; and CoLA (Warstadt et al., 2018), which annotates sentences for grammatical acceptability;

Each dataset was split into train, dev, and test sets, keeping the original split, if exists, and otherwise applying a $70\%/10\%/20\%$ split, respectively. Since we are interested in limited labeled data settings, we used the dev sets only for optimizing the baseline algorithms, denoted by Path-1 and Path-2 in Fig. 1. The dev sets were *not* used in the algorithmic flows denoted by Path-3 and Path-4 in that figure. Since MLM inter-training is computationally demanding, for DBpedia, AG's News, Yahoo! Answers and CFPB, which are relatively large datasets, we limit the sizes of the train/dev/test sets to $15K/3K/3K$ instances respectively, by randomly sampling from each set[3]. Links to all datasets are provided in Appendix §A.

## 3.2 EXPERIMENTAL SETUP

In our main set of experiments we compare the performance of fine-tuning BERT-based models over a target task, for different settings of intermediate training. We consider four BERT-based settings, as described in Section 2 and in Figure 1. Two baselines – (i) BERT, where no intermediate training is applied, and (ii) $BERT_{IT:MLM}$, where MLM is used as the inter-training task; and two settings that rely on clustering – (1) $BERT_{IT:CLUST}$, where predicting cluster labels is used for inter-training, and (2) $BERT_{IT:MLM+CLUST}$, which combines the two intermediate tasks. In addition, we include several non-BERT baseline settings which rely on simpler models.

**Training samples:** For each setting, the final model training for the target task (fine-tuning, in the case of BERT-based models) is performed, per dataset, for training budgets varying between $64$ and $1024$ labeled examples. For each data size $x$, the experiment is repeated 5 times; each repetition representing a different sampling of $x$ labeled examples from the train set. Sampling of training examples is matched between all settings. That is, for a given dataset and train size the final training run for all settings is done with respect to the same 5 samples of labeled examples [4]. Results per repetition appear in Fig. 4 of Appendix §B.

**Inter-training:** Intermediate training, when done, was performed over the full train set for each dataset (ignoring instances' labels). We studied two implementations for the clustering task: k-means (Lloyd, 1982) and sequential Information Bottleneck (sIB) which is known to obtain better results in practice (Slonim et al., 2002) and in theory (Slonim et al., 2013). Based on initial experiments, and previous insights from works in the computer vision domain (Yan et al., 2020) we opted for a relatively large number of clusters, and rather than optimizing the number of clusters per dataset, set it to $50$ for all cases.[5] The k-means baseline was run over GloVe (Pennington et al., 2014) representations following word stemming. We used a publicly available implementation of sIB[6] with its default configuration (i.e., 10 restarts and a maximum of 15 iterations for each single run). For the clustering, we used Bag of Words (BOW) representations on a stemmed text with the default vocabulary size (which is defined as the 10k most frequent words in the dataset). Our results indicate that inter-training with respect to sIB clusters consistently led to better results in the final performance on the target task, compared to inter-training with respect the clusters obtained with k-means. Hence, in the main paper we present results using the sIB implementation only, while results including k-means are discussed in Appendix §C. We also considered inter-training only on

---

[3]We verified that relying on this limited set has no significant impact on the performance of $BERT_{IT:MLM}$ and $BERT_{IT:CLUST}$ compared to using the full dataset. The results are omitted for brevity.

[4]Note for reviewers: 11 of the total 400 samplings did not cover all target classes, and fine-tuning was not performed, hence results reflect an average over less than 5 repetitions. For 20 newsgroups, budget of 64 all 5 samples did not cover all classes and hence this data point is not presented. We plan to re-sample for completing the missing data points.

[5]Setting the number of clusters to be equal to the number of classes resulted in inferior accuracy. In addition, we note that in practice one may not know how many classes truly exist in the data, so this parameter is not necessarily known in real-world applications.

[6]https://github.com/IBM/sib

representative examples of clustering results, filtering a given amount of outlier examples. Filtering did not improve results (data not shown).

Note that the run time of the clustering algorithms is negligible, and only takes a few seconds. The run time of the fine-tuning step of the inter-training task takes five and a half minutes for the largest train set (15k instances) on a Tesla V100-PCIE-16GB GPU.

**BERT hyper-parameters:** The starting point of the various inter-training settings is the $\text{BERT}_{\text{BASE}}$ model (110M parameters). BERT inter-training and fine-tuning runs were all performed using the Adam optimizer (Kingma & Ba, 2015) with a standard setting consisting of a learning rate of $3 \times 10^{-5}$, batch size 64, and maximal sequence length 128.

The number of training epochs varies by setting. For the baselines of BERT and $\text{BERT}_{\text{IT:MLM}}$, fine-tuning was performed over 15 epochs, selecting the best epoch based on accuracy over the dev set. Since in a practical limited annotations budget setting one cannot assume that a labeled dev set is available, in our $\text{BERT}_{\text{IT:CLUST}}$ and $\text{BERT}_{\text{IT:MLM+CLUST}}$ settings we did not use a dev set, and fine-tuning was arbitrarily set to be over 10 epochs. For inter-training over the clustering results we used a single epoch for two reasons. First, loosely speaking, additional training over the clusters may drift the model too far towards learning the partition into clusters, which is an auxiliary task in our context, and not the real target task. Second, from the perspective of a practitioner, single epoch training is preferred since it is the least demanding in terms of run time. For MLM inter-training we used 30 epochs with a replication rate of 5. We note that using fewer epochs yielded similar results. As this may be of interest for other researchers, we provide the results in Appendix §D.

**Non-BERT baselines:** As additional baselines, the same training samples were also used to train multinomial Naive Bayes (NB) and linear Support Vector Machine (SVM) classifiers, using either Bag of Words (BOW) or GloVe (Pennington et al., 2014) representations. For GloVe, a text is represented as the average GloVe embeddings of its tokens. This yielded four additional baseline settings: $\text{NB}_{\text{BOW}}$, $\text{NB}_{\text{GloVe}}$, $\text{SVM}_{\text{BOW}}$ and $\text{SVM}_{\text{GloVe}}$.

## 4 RESULTS

Figure 2 depicts the classification accuracy for the different experimental settings in each of the datasets, for varying labeling budgets. The $\text{SVM}_{\text{GloVe}}$ baseline outperformed the other baseline settings, therefore, we only include it in this figure; Full results for all NB and SVM settings can be found in Appendix §E.

Evidently, in all 6 topical datasets, $\text{BERT}_{\text{IT:CLUST}}$ and $\text{BERT}_{\text{IT:MLM+CLUST}}$ clearly outperform BERT and $\text{BERT}_{\text{IT:MLM}}$ in the small labeled data regime, where the gain is most prominent for the smallest labeled data examined – when only 64 labeled examples are available – and gradually diminishes as more labeled samples are added. In the remaining 4 non-topical datasets the clustering inter-training does not appear to confer a similar benefit. Nevertheless, even in these datasets, $\text{BERT}_{\text{IT:CLUST}}$ results are typically comparable to the baseline algorithms.

| Dataset | BERT accuracy | BERT$_{\text{IT:CLUST}}$ accuracy | Gain | Error reduction |
|---|---|---|---|---|
| Yahoo! Answers | 24.7 | 45.9 | 86% | 28% |
| DBpedia | 42.2 | 67.0 | 59% | 43% |
| CFPB | 22.1 | 27.5 | 24% | 7% |
| 20 newsgroup | 19.7 | 47.2 | 139% | 34% |
| AG's News | 73.3 | 80.7 | 10% | 28% |
| ISEAR | 19.0 | 29.0 | 52% | 12% |
| avg. topical | | | **62%** | **25%** |
| SMS spam | 98.0 | 98.2 | 0% | 10% |
| Subjectivity | 90.7 | 91.0 | 0% | 3% |
| Polarity | 67.7 | 67.0 | -1% | -2% |
| CoLA | 69.5 | 66.0 | -5% | -11% |
| avg. non-topical | | | **-1%** | **0%** |

Table 2: Classification accuracy for BERT and $\text{BERT}_{\text{IT:CLUST}}$ when using 64 samples for fine-tuning, the accuracy gain relative to BERT's accuracy and the reduction in error (1-accuracy) relative to the BERT error.

Table 2 depicts the results from Figure 2, focusing on the practical use case of a minimal budget of 64 samples for fine-tuning. As is evident, $\text{BERT}_{\text{IT:CLUST}}$ confers a significant benefit in accuracy (62% accuracy gain, 25% error reduction on average), when considering the topical datasets.

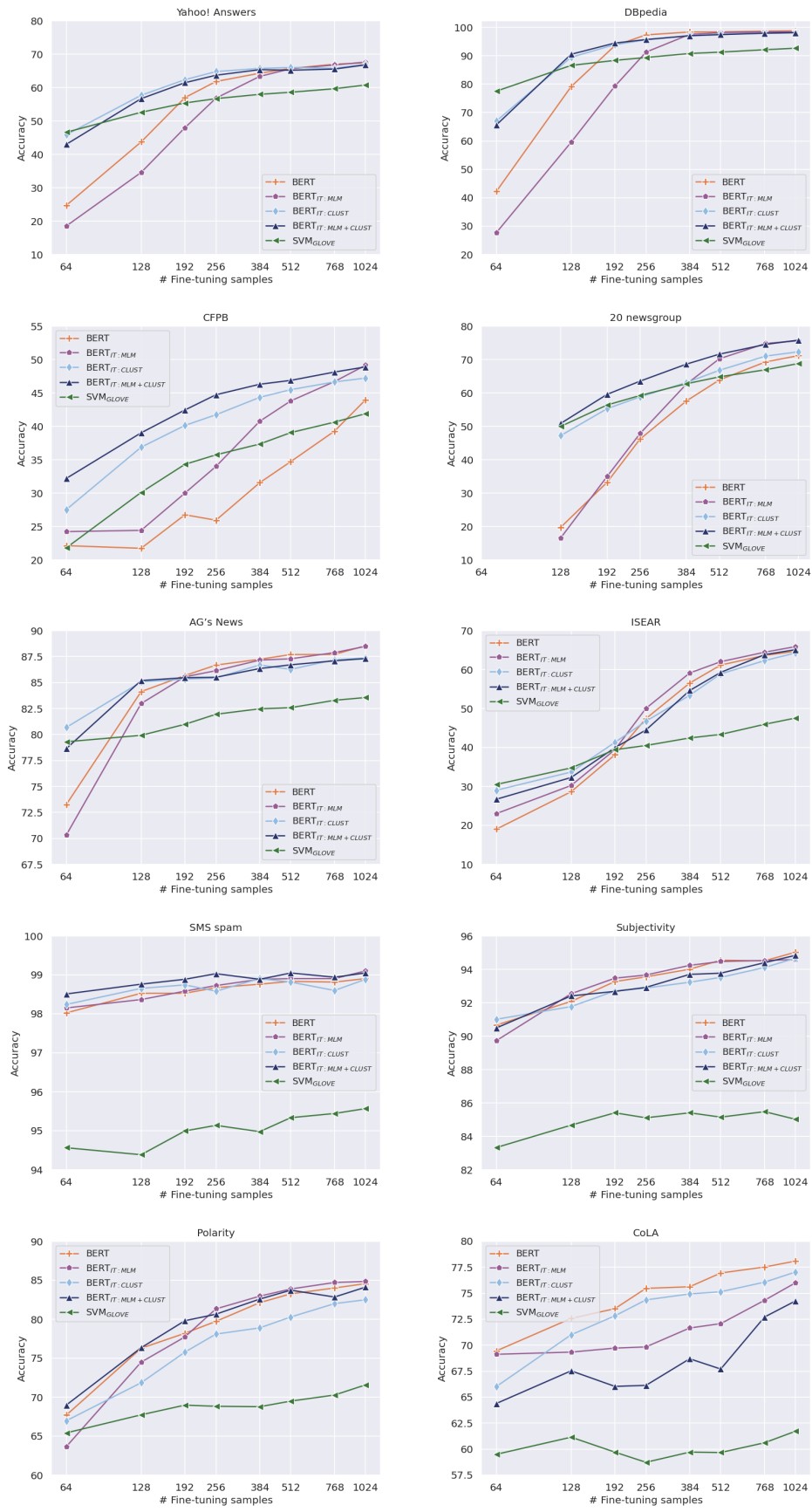

Figure 2: Classification accuracy over the test set for the different experimental settings versus the number of labeled samples used for fine-tuning (log scale). Each point is the average of five repetitions.

| Train size | 64 | 128 | 192 | 256 | 384–512 | 768 | 1024 |
|---|---|---|---|---|---|---|---|
| BERT$_{\text{IT:CLUST}}$ † vs. BERT | $2\times10^{-4}$† | $3\times10^{-4}$† | $2\times10^{-3}$† | $--$ | $--$ | $--$ | $--$ |
| BERT$_{\text{IT:CLUST}}$ † vs. BERT$_{\text{IT:MLM}}$ ‡ | $2\times10^{-4}$† | $2\times10^{-5}$† | $8\times10^{-6}$† | $2\times10^{-3}$† | $--$ | $6\times10^{-4}$‡ | $1\times10^{-4}$‡ |

Table 3: Paired t-test p-values (after Bonferroni correction) of classification accuracy for BERT$_{\text{IT:CLUST}}$ compared to BERT (top row) and to BERT$_{\text{IT:MLM}}$ (bottom row). Symbols indicate the setting which achieved better performance. $--$ denotes insignificant results ($p \geq 0.05$).

Table 3 includes statistical significance analysis. We perform paired t-tests to compare BERT$_{\text{IT:CLUST}}$ with BERT and BERT$_{\text{IT:MLM}}$, pooling together all datasets and repetitions for a given labeling budget. For a budget up to 192 examples, BERT$_{\text{IT:CLUST}}$ significantly outperforms both BERT and BERT$_{\text{IT:MLM}}$, expressing the robustness of the proposed approach.

BERT$_{\text{IT:CLUST}}$ also outperforms the non-BERT baselines. For a few topical datasets, SVM$_{\text{GloVe}}$ is on par with or slightly superior to BERT$_{\text{IT:CLUST}}$ for the smallest train budget. Overall, when considering the entire range of fine-tuning sample sizes, its results are inferior to our approach and it often performs quite poorly.

The different behavior between the topical datasets versus the non-topical datasets highlights that the BERT$_{\text{IT:CLUST}}$ configuration we considered seems most valuable for topical tasks. Specifically, we note that both clustering algorithms we examined rely on BOW representations and correspondingly are better suited for topical data in finding a partition that reasonably approximates the true hidden partition of the data, according to the class labels. Therefore, we try using the BERT [CLS] token representation as a non-BOW representation. While it seems somewhat better than BOW on the non-topical Polarity dataset, it is not consistently better on the other non-topical datasets, and it is mostly worse on the topical datasets (see Appendix §C). Future work should examine ways to make BERT$_{\text{IT:CLUST}}$ better suited for non-topical datasets, possibly by capturing stylistic distinctions.

The performance gains of BERT$_{\text{IT:CLUST}}$ over BERT$_{\text{IT:MLM}}$ suggest that the potential benefits of BERT$_{\text{IT:CLUST}}$ do not consist merely of adapting the model to the characteristics of the target class corpus; rather, it appears that inter-training on top of the clustering results carries additional benefit. A natural explanation is that the pseudo-labels obtained via the clustering partition are informative with regards to target task labels. To quantify this intuition, in Figure 3 we depict the Normalized Mutual Information (NMI) between cluster labels and the target task labels, calculated over the entire training set, versus the gain of using BERT$_{\text{IT:CLUST}}$ – reflected as the reduction in classification error rate between BERT and BERT$_{\text{IT:CLUST}}$ – at the extreme case of $64$ fine-tuning samples. Evidently, in datasets where the NMI is around $0$, BERT$_{\text{IT:CLUST}}$ does not confer a clear benefit; conversely, where the NMI is relatively high, the performance gains are pronounced as well.

Finally, since the partition obtained via unsupervised clustering is often informative for the target class labels, we examine whether it can be utilized directly, as opposed to as pseudo-labels for BERT inter-training. To that end, we applied a simple heuristic. Given a labeling budget, we divide it across all clusters, and use the budget per cluster to reveal the labels of a random sample of examples in that cluster, and then identify each cluster with the most dominant label found in it. Next, given a new test example, we assign it with the label associated with its nearest cluster. In Appendix §B and Figure 4 we provide full details, and share the performance of this simple baseline. While such a rudimentary classifier can be surprisingly effective, especially where the NMI is high and the labeling budget is low, it is generally not on par with BERT$_{\text{IT:CLUST}}$ performance.

Taken together, our analyses suggest that in topical datasets, where labeled data is scarce, the pseudo-labels generated via simple text clustering techniques can be leveraged by BERT for inter-training, to provide a better starting point of the model towards its final fine-tuning for the target task.

## 5 RELATED WORK

In our work, we transfer the pretrained BERT (Devlin et al., 2018) model to a new domain with little data. There is a whole field studying how to transfer models across domains, namely, transfer

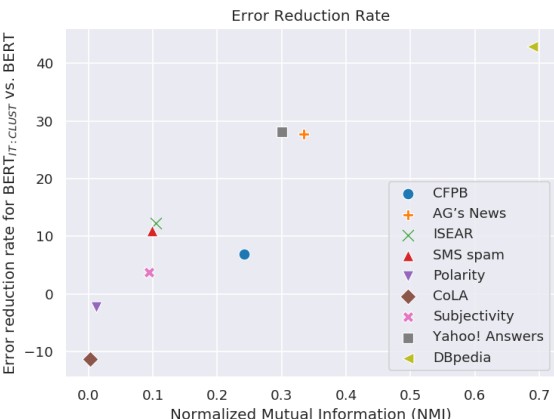

Figure 3: Improvement by BERT$_{\text{IT:CLUST}}$ vs Normalized Mutual Information per dataset. x-axis: Normalized Mutual Information between the cluster labels and the class labels, calculated over the entire train set. y-axis: The reduction in classification error (percentage) of inter-training using BERT$_{\text{IT:CLUST}}$ relative to no inter-training, when using 64 samples in the fine-tuning phase.

learning. It suggests methods such as pivoting (Ziser & Reichart, 2018), weak supervision (Shnarch et al., 2018), and adversarial transfer (Cao et al., 2018).

In Computer Vision, pretrained models are often learnt by image clustering tasks (Caron et al., 2018). In Natural Language Processing, however, clustering was mainly used for non-transfer scenarios. Ball (2019) relies on pretrained embeddings to cluster labeled and unlabeled data. Then, they fill the missing labels to augment the training data of any classifier. Clustering itself was improved by combining small amounts of data (Torres & Vaca, 2019; Wang et al., 2016).

Pretrained models improved state-of-the-art in many tasks (Nogueira & Cho, 2019; Ein-Dor et al., 2020) and they are especially needed and useful in low resource and limited labeled data settings (Lacroix et al., 2019; Wang et al., 2020a; Chau et al., 2020). There are many suggestions to improve such models, including larger models (Raffel et al., 2019), changes in the pretraining tasks and architecture (Yang et al., 2019), augmenting pretraining (Geva et al., 2020) or improving the transfer itself (Valipour et al., 2019; Wang et al., 2019b; Sun et al., 2019; Xu et al., 2020). Two findings on pretraining support our hypothesis on the intermediate task, namely, classification surpass MLM. Some pretraining tasks are better than others (Lan et al., 2020; Raffel et al., 2019) and supervised classification as additional pre-training improves performance (Lv et al., 2020; Wang et al., 2019a; Pruksachatkun et al., 2020). All these works aim to improve the performance upon transfer, making it more suitable for any new domain. In contrast, we focus on the improvement given the domain.

With a transferred model, one can further improve performance with domain-specific information. For example, utilizing metadata (Melamud et al., 2019), training on weakly-supervised data (Raisi & Huang, 2018) or multitasking on related tasks concurrently (Liu et al., 2019a). Given no domain-specific information, it was suggested to further pretrain on unlabeled data from the domain (Whang et al., 2019; Xu et al., 2019; Sung et al., 2019; Rietzler et al., 2020; Lee et al., 2020; Gururangan et al., 2020). This, however, is sometimes unhelpful or even hurts results (Pan, 2019). We replicate this finding for cases in which labeled data is scarce (see Section §4).

Transferring a model and retraining with paucity of labels is often termed few shot learning. Few shot learning is used for many language related tasks such as named entity recognition (Wang et al., 2020b), relation classification (Hui et al., 2020) and parsing (Schuster et al., 2019). There have also been suggestions other than fine-tuning the model. Koch (2015) suggests to rank the similarity between examples with Siamese networks. Vinyals et al. (2016) rely on memory and attention to find neighboring examples and Snell et al. (2017) search for prototypes to compare to. Ravi & Larochelle (2017) don't define in advance how to compare the examples. Instead, they meta-learn how to train the few shot learner. These works addressed the image classification domain, but they

supply general methods which are used, improved and adapted on language domains (Geng et al., 2019; Yu et al., 2018).

In conclusion, separate successful practices foreshadow our findings: Clustering drives pre-training on images; supervised classification aids pre-training; and training on unlabeled domain examples is helpful with MLM.

## 6 DISCUSSION

We presented a simple approach to improve BERT-based models for topical text classification. Specifically, we show that inter-training BERT over pseudo-labels generated via unsupervised sIB clustering results in a model that represents a better starting point for the final fine-tuning over the target task at hand. Thus, our analysis suggests that BERT can leverage these pseudo-labels, namely that there exists a beneficial interplay between the proposed inter-training and the fine-tuning stage. Our results show that this approach yields a consistent significant boost in BERT accuracy over topical data when labeled data is scarce.

We opted here for a practically oriented approach, which we do not claim to be optimal. Rather, the success of this approach suggests various directions for future work. In particular, several theoretical questions arise, such as what determines the success of the approach in a given dataset; understand the potential synergistic effect of using BOW-based clustering for inter-training BERT representations; could more suitable partitions be acquired by exploiting additional embedding space and/or more clustering techniques; co-training (Blum & Mitchell, 1998) methods, and more.

On the practical side, while in this work we focused on inter-training over 50 clusters for a single epoch, more work is needed to determine how to tune such hyper-parameters. In addition, one may consider using the labeled data available for fine-tuning as anchors for the intermediate clustering step, which we have not explored here.

Another point to consider is the nature of the inter-training task. Here, we examined a multi-class setup where BERT is trained to predict one out of $n_c$ cluster labels. Alternatively, one may consider a binary inter-training task, where BERT is trained to determine whether two samples are drawn from the same cluster or not.

Finally, the focus of the present work was on improving BERT performance for text classification. In principle, inter-training BERT over clustering results may be valuable for additional downstream target tasks, that are similar in spirit to standard text classification. Examples include recent work on Key-Point Analysis (Bar-Haim et al., 2020) and the task of Textual Entailment (Dagan et al., 2013). The potential value of our approach in these and other cases is left for future work.

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

## A DATASETS

In this paper we used 10 datasets. Table 1 provides details about their split into train, dev, and test sets. For each set its size and the prior of the target class is presented.

Links for downloading the datasets:

**Polarity:** `http://www.cs.cornell.edu/people/pabo/movie-review-data/`.

**Subjectivity:** `http://www.cs.cornell.edu/people/pabo/movie-review-data/`.

**CFPB:** `https://www.consumerfinance.gov/data-research/consumer-complaints/`.

**20 newsgroups:** `http://qwone.com/~jason/20Newsgroups/` We used the version provided by scikit: `https://scikit-learn.org/0.15/datasets/twenty_newsgroups.html`.

**AG's News, DBpedia and Yahoo! answers:** We used the version from: `https://pathmind.com/wiki/open-datasets` (look for the link *Text Classification Datasets*).

**SMS spam:** `http://www.dt.fee.unicamp.br/~tiago/smsspamcollection/`

**ISEAR:** `https://www.unige.ch/cisa/research/materials-and-online-research/research-material/`.

**CoLA:** `https://nyu-mll.github.io/CoLA/`

## B sIB-BASED CLASSIFIER

As described in section 4, we experimented with building a rudimentary classifier that utilizes only the sIB clustering results and the labeling budget. We estimate the most common label for each cluster by labeling some of its instances, using a given labeling budget. The budget is distributed

among the clusters relative to their size, while ensuring that at least one instance from each of the 50 clusters is labeled. Then, each cluster is assigned the majority label from its labeled instances. Test examples are classified according to the label associated with the nearest cluster. Results for this setting are depicted in Fig. 4. Comparing these results to the BERT-based approaches reveals that clustering alone is not sufficient.

## C  EXAMINING ADDITIONAL CLUSTERING TECHNIQUES

In addition to sIB over BOW (denoted $BERT_{IT:CLUST}$), we evaluated three more configurations for the clustering intermediate task; K-means over GloVe representation, K-means over BERT CLS, and Hartigan's K-means (Slonim et al., 2013) over GloVe. For BERT CLS, for each input text we take the representation of the [CLS] token from the last hidden layer of the $BERT_{BASE}$ model.

Results are shown in Fig. 5. When comparing to sIB, on eight out of ten occasions sIB over BOW outperforms the other clustering configurations.

In initial trials, sIB was also showing better results as a clustering method, which was the reason we used it rather than K-means as our main intermediate tasks. Because of the change in quality, there is not enough evidence to say whether sIB is better as a pretraining task due to clusters that are more related to the target task or due to some characteristics of the clusters themselves. Such different characteristics may arise, for example, from the ways clusters are chosen. K-means relies on Geometric notions such as distance to cluster while sIB on information theoretic notions.

## D  THE EFFECT OF THE NUMBER OF MLM EPOCHS

We tested several options for the number of epochs of intertraining using MLM. The results in Fig. 6 don't show a clear choice, although all cases are not consistently improving over the baseline. We thus chose 30 epochs, as MLM scores seemed somewhat more favourable.

## E  NON-BERT BASELINES

The results of the $NB_{BoW}$, $NB_{GloVe}$, $SVM_{BoW}$ and $SVM_{GloVe}$ baselines are shown in Figure 7.

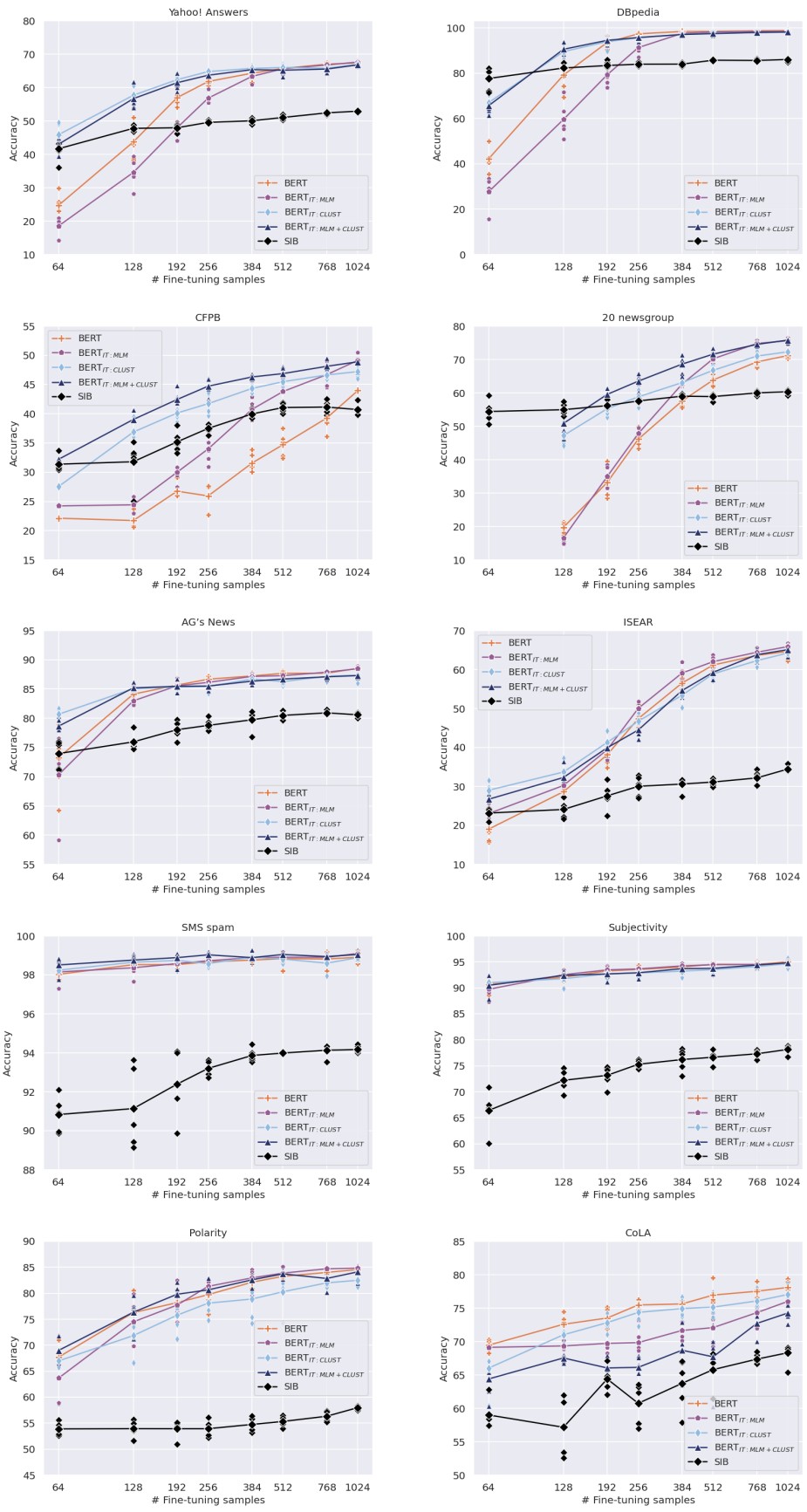

Figure 4: Classification accuracy over the test set for the different experimental settings versus the number of labeled samples used for fine-tuning. Each point in the line is the average of five repetitions. The repetitions themselves are shown. The results of the sIB-based classifier are also shown. X axis denotes the budget for training in log scale, and Y accuracy of each model.

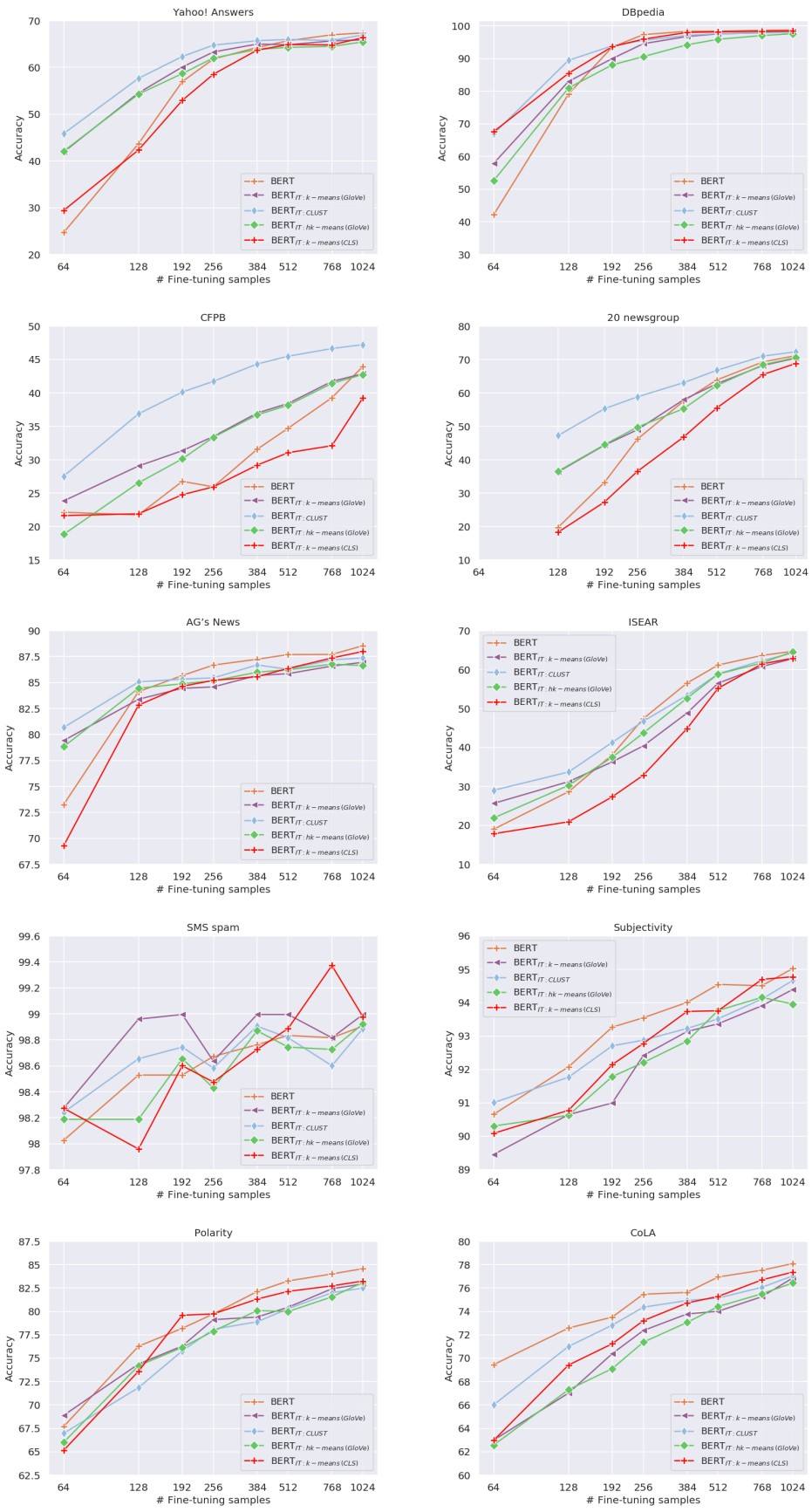

Figure 5: Comparing various clustering configurations: (1) K-means over Glove, (2) K-means over CLS, (3) Hartigan's K-means over Glove, (4) sIB over BOW, and (5) no intermediate task. Each point is the average of five repetitions. X axis denotes the budget for training in log scale, and Y accuracy of each model.

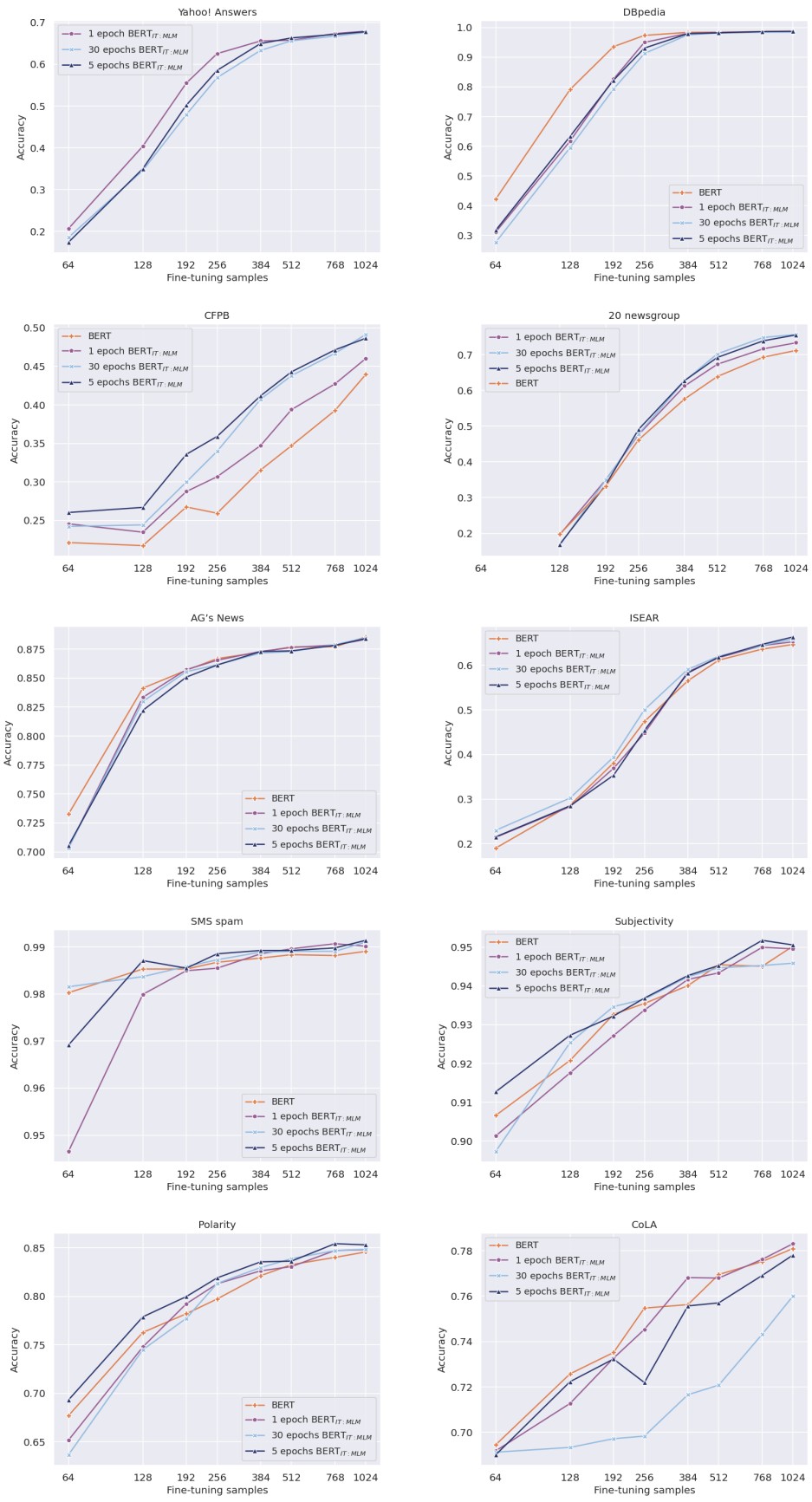

Figure 6: Evaluating the effect of different number of MLM epochs. Each point in the line is the average of five repetitions. X axis denotes the budget for training in log scale, and Y accuracy of each model.

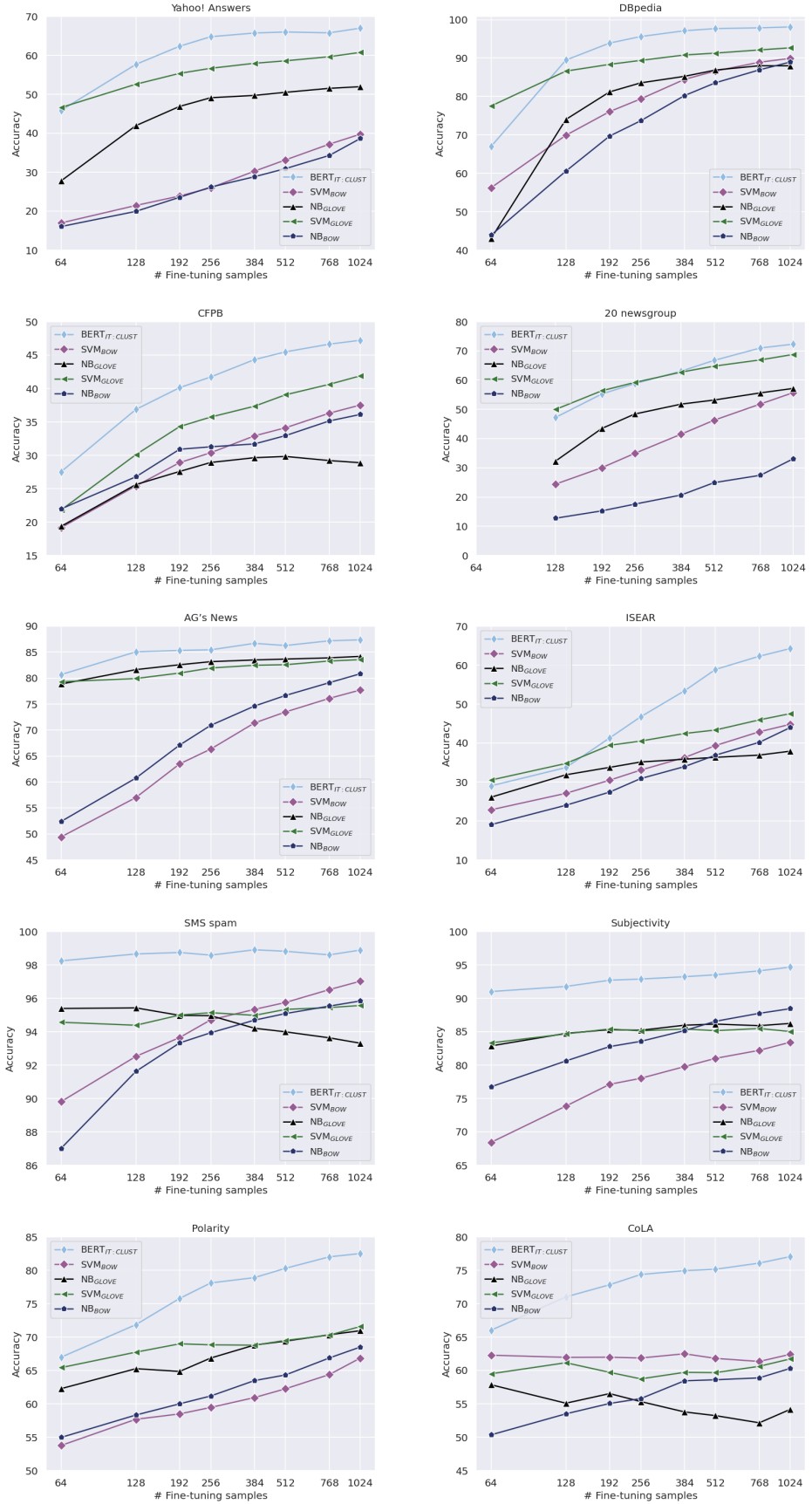

Figure 7: Comparing non-BERT baselines and the BERT$_{IT:CLUST}$ setting. Each point is the average of five repetitions. X axis denotes the budget for training in log scale, and Y accuracy of each model.

