# OpenReview forum: "Cluster & Tune: Enhance BERT Performance in Low Resource Text Classification"
_ICLR.cc/2021/Conference — Reject_

### Official Review · AnonReviewer1 · 2020-10-25
**An interesting paper that lacks context and baselines**

**Rating:** 6
**Confidence:** 4

**Review:**

This paper proposes a novel domain/task adaptation procedure for BERT-style language models (LMs). Inspired by computer vision, the authors propose to specialize LMs to a particular domain and task with an intermediate clustering task. A large unlabeled dataset is clustered to generate pseudo-labels then a BERT model is specialized to predict the cluster id of each example. More precisely, the authors use GloVe embeddings (bag-of-word representations) and the sIB clustering algorithm with a constant number of clusters (50).

The additional clustering task is shown to help for low-resource topic classification. In the paper setup, the whole dataset (up to 15000 examples) is used for clustering (without the labels), while the fine-tuning phase relies on a small subset of the training data. The authors show nice gains when using 64 examples for fine-tuning. Graphs show that the gains are lower as more fine-tuning data is used, and usually disappear with more that 192 to 384 examples. The authors go on showing that their approach works best when the cluster ids correlate well with the original labels.

Overall the paper is clear and well-written. It proposes an interesting and, to the best of my knowledge, novel idea.

Concerns:
- Some claims feel overly broad. I believe the claims should be limited to improving low-resource/few-shot *topic* classification tasks, not text classification in general, which is especially unclear in the title and abstract.
- While the domain transfer literature is well introduced, no low-resource/few-shot literature is mentioned, making it hard to relate the results to similar research (although to be fair, it is true that CV has much more literature on this topic than NLP).
- There is little discussion on the motivation and practicality of this approach, which makes the results seem almost anecdotic. If the motivation is to improve few-shot learning, the relevant literature should be introduced and compared against. If the motivation is topic classification, the practicality of this work seems limited: since a larger dataset is available for clustering, manually labeling a few hundred examples (human expert, MTurk) feels more promising. For example, BERT with 256 examples yields better results on most datasets than BERT:CLUST with 64 examples.
- Several ideas mentioned as future work seem relatively straightforward to try and could have been interesting for inclusion in this paper.

Questions:
- The correlation between cluster ids and task labels seems critical for the approach to work. Did the authors try using a number of clusters equal to the number of labels? If so, how did that change the results?
- The vocabulary size used for clustering (10000 word stems) seems low. What are the reasons for choosing that number? Also, is the vocabulary task-dependent?
- Why did the authors use a BoW approach for clustering? For example, why not embed each example with BERT (which should capture topics better than GloVe-BoW) using the output [CLS] vector?

---

> ### Author Response · Authors · 2020-11-23
> **updating our claim to topical text classification, addressing the other concerns, and answering the questions**
>
> Thank you for your careful analysis of our paper. We accept your comment that we should better reflect the fact that our method was eventually found most useful for *topical* text classification. Per your suggestion, we have updated the title, the abstract, and text.
>
> In addition, we have added a paragraph on few-shot learning literature in the related work section.
>
> Regarding your comment on the motivation of this work. Our motivation is not improving few-shot learning, rather it is improving real-world scenarios of topic classification where there is a strict limit on the amount of annotation that can be collected. There are many cases in which the data cannot be sent for labeling by the crowd (e.g., due to confidentiality of the data, or
> the need for a very specific expertise). Often, the availability of experts for annotation is very limited and even as much as 250 labeled examples are too much to expect. In the paper, in the results figures, the x axis goes to 1024 in order to show where the different methods reach the same accuracy. However, our focus in this paper is the left-hand side of the x axis, where the amount of labeled examples is very limited. This area reflects a real need that we, and many others, are facing. We have updated the paper to better present our motivation.
>
> Per your suggestion to try to include something from the future work in this paper - we have implemented one direction of our future work. In the original discussion section, we raise the following question: "could more suitable partitions be acquired by changing embedding space?". We have tested k-means over the CLS token of BERT as well as classifying with SVM over a representation based on Glove. Both methods do not yield better performance (see Figures 5 and 7).
>
> Questions:
>
> * "Did the authors try using a number of clusters equal to the number of labels?" Yes, we have, and the results were consistently inferior.
> Following your question, we now specifically mention this point in the footnote in section 3.2, experimental setup. In addition, we note that in practice one may not know how many classes truly exist in the data, so this parameter is not necessarily known in real-world applications. Furthermore, we wanted to show that the gain is robust, hence used a single configuration with 50 clusters across all datasets.
>
>
> * "The vocabulary size used for clustering seems low. What are the reasons for choosing it? Is the vocabulary task-dependent?"
> We used the default size of the open source code we used, which is defined as the 10k most frequent words in the dataset (we added this to the experimental setup section). Thus, the vocabulary size is *dataset* dependent. We have not tried to optimize the vocabulary size parameter, though potentially this could result in better performance.
>
> * "Why did the authors use a BoW approach for clustering?  For example, why not embed each example with BERT ... using the output [CLS] vector?"
> Your suggestion to use the CLS is indeed a good direction to explore. We have therefore implemented it and have added clustering over BERT's [CLS] to our results - see Figure 5. As can be seen in the figure, the BOW approach performs better.

---

### Official Review · AnonReviewer2 · 2020-10-28
**A good way to fine tune BERT for topical text classification with fewer labels.**

**Rating:** 6
**Confidence:** 3

**Review:**

Quality

The paper proposed using unsupervised clusters to help boost BERT performance in text classification tasks that lack labels. The method make good sense. Experiments well designed and showed clear advantage.

Clarity

The paper is well written. I have no trouble following all details.

Originality

It might be the first for BERT, but using unsupervised clustering to help classification seems an old topic in NLP.

Significance

It's less significant. First, it limits to topical classification. Second, it fits the scenario with more data for clustering (thus not truly low resource) but just too few labels. Performance gain diminishes quickly as number of labels get over a few hundreds.

Why not higher rating? The method seems too intuitive from NLP research. It's about BERT but the idea is not that novel. It's only helping topical cases with quick diminishing gains.

Why not reject? Paper is well written, evaluation done thoroughly and showed good improvement. Authors analyzed the result in a useful way.

Detailed comments:
1) The authors should be careful about the "low resource" claim, as we still need fair amount of data, unlabeled, for the clustering.
2) Did you use training set for clustering, without considering their labels? Seems so but not clarified in the paper.
3) Why use accuracy, not P/R as the metric? are all test sets balanced?
4) For Figure 3, consider using shapes or in-figure labels. Color dots are hard to read when we have many labels, and not friendly to color-blind people either.

---

> ### Author Response · Authors · 2020-11-23
> **replacing the term "low resource" and why performance gain in the realm of less than a few hundreds labels matters**
>
> Thank you for your careful analysis of our paper. We accept your comment about the usage of the term "low resource" and have changed the title, abstract and text to avoid this term, replacing it with the more accurate term of "limited amount of labeled data".
>
> You mentioned that the fact that "performance gain diminishes quickly as number of labels get over a few hundreds" is a reason for not giving a higher rating. We'd like to better present our motivation, which lies exactly in the realm of less than a few hundreds labeled examples.
> In many real-world scenarios of topic classification there is a strict limit on the amount of annotation that can be collected. There are many cases in which the data cannot be sent for massive labeling by the crowd (e.g., due to confidentiality of the data, or the need for a very specific expertise). Often, the availability of experts for annotation is very limited and even as many as a few hundreds labeled examples are too much to expect. In the paper, in the results figures, the x axis goes to 1024 in order to show where the different methods reach a similar accuracy. However, our focus in this paper is the left-hand side of the x axis, where the amount of labeled examples is very small. This area reflects a real need that we, and many others, are facing. We have updated the paper to better present our motivation.
>
> You asked, "Did you use training set for clustering, without considering their labels?" The answer is yes, and we added a note in the Inter-training paragraph of section 3.2.
>
> You also asked, "Why use accuracy, not P/R as the metric? are all test sets balanced?"
> We opted for using accuracy in order to be aligned with the standard measures used in the leader boards of most of the datasets used. See for example the leader boards in https://paperswithcode.com/area/natural-language-processing which report accuracy (or error, which is 1-accuracy). The datasets are indeed balanced, apart of two, SMS-Spam and Cola, for which F1 is indeed a more natural measure (both are binary classification tasks with a positive class which is significantly less than half of the data), but we wanted to keep a consistent measure. We did verify that the qualitative results with respect to the different algorithms in the imbalanced datasets do not change when using a weighted F1. If you think this is an important point, we can add it in the paper.
>
> Finally, Figure 3 was improved per your suggestion.

---

### Official Review · AnonReviewer3 · 2020-10-28
**Well-written, comprehensive experiments and analyses. Will be applicable for any topical dataset without a plenty of labels.**

**Rating:** 8
**Confidence:** 3

**Review:**

This paper proposed an inter-training framework of BERT, an unsupervised training method applied between the conventional pre-training method via Masked Language Models and the fine-tuning method by using labeled data in the target task. The proposed inter-training methods are an unsupervised method by first running clustering with BoW features for the target task and then fine-tuning by training BERT over pseudo-labels generated by clustering results. This inter-training framework significantly improves prediction accuracies especially when the task is topical, i.e., the task to classify texts based on a high-level distinction related to what the text is about, and the labeled data is scarce.

This paper is well-written. The motivation is reasonable and the proposed methods make sense. Experiments are comprehensive and analyses are well designed. The contribution of this paper is obviously above the ICLR borderline.

I think it is better if the authors mention the computational cost of the inter-training framework in detail. While the computational cost of the clustering is negligible, the authors did not mention the computational cost of the fine-tuning in the inter-training framework. Compared with the final fine-tuning process, it is better to clarify how much additional cost we need to perform the inter-training method. Moreover, it is better if the authors mention the effect of the size of unlabeled data in the inter-training method. Is more unlabeled data improve target metrics a lot, or are hundreds of texts sufficient for the inter-training method. It is better if the authors show the plot by changing the size of the unlabeled data used for the inter-training method.

---

> ### Author Response · Authors · 2020-11-23
> **runtime answer and we will add a plot of the effect of the size of the unlabeled data**
>
> Thank you for your careful analysis of our paper and your positive and encouraging review.
>
> The run time of the fine-tuning step of the inter-training task takes five and a half minutes for the largest train set (15k instances) on Tesla V100-PCIE-16GB GPU. We added this information to the paragraph discussing run time (at the end of the Inter-training paragraph in section 3.2).
>
> Per your suggestion, we will add, for the final version, a plot of the effect on accuracy of the size of the unlabeled data used for the inter-training method.

---

### Official Review · AnonReviewer4 · 2020-10-29
**Simple BoW related baselines are missing**

**Rating:** 3
**Confidence:** 4

**Review:**

The paper proposes to use a simple intermediate task - clustering - to improve the generalization ability of BERT in low resource text classification settings. The idea is simple. We can perform clustering using BoW representations to generate pseudo labels, which will be used to fine-tune the pre-trained BERT model. The results show that the proposed method is effective in topical classification problems.

Pros:
 - the idea is very simple and we can easily adapt the idea to different classification tasks.
- the results are promising, especially on topical classification problems.

cons:
- My main doubt about this paper is that I think the improvements in the topical classification problems were rooted in the fact that the authors used the "BoW representations" to conduct the clustering. The BoW representations are known to be more effective for topical classification problems compared to BERT-based sentence representations. So what the authors did maybe actually infuse/distill the BoW knowledge into the BERT representations.
- I suggest the authors include a set of BoW representation based baselines. E.g., simple BoW based SVM model, average of GloVe embeddings of BoW representations based classification.

---

> ### Author Response · Authors · 2020-11-23
> **We added four BOW related baselines to the paper - results were overall inferior to our approach**
>
> Thank you for your careful analysis of our paper, and especially for highlighting the need for comparing our results to simple BOW related baselines. We agree that such an evaluation is imperative to better understand the potential of our proposed method. Correspondingly, we tested 4 such baselines, only one of which resulted in relatively decent performance. The results of this method are now added to Figure 2, along with a corresponding discussion. As can be seen in the figure, even for this most successful baseline, the results were overall inferior to our approach. Thus, while we agree with the reviewer that our approach perhaps benefits from infusing BOW knowledge into the BERT representations, our updated results suggest that in practice BERT-IT-CLUST should still be preferred over simpler alternatives, even for topical datasets, and certainly for non-topical ones. A better understanding of the potential synergistic effect of using BOW-based clustering for inter-training BERT representations merits further investigation, as we now point out in the Discussion.
>
> *Further details*:
>
> As baselines, we examined SVM and Naive Bayes, over the explicit BOW representation and over Glove representation (where each text is represented based on the average Glove embeddings of its words). A comparison of all BOW baselines to BERT is presented in the new Figure 7 in the Appendix. For three of these baselines, the results were consistently lower than the results reported in our original submission, typically by a very significant margin. The only baseline with relatively decent performance was SVM-Glove, hence the results of this method are now added to Figure 2, along with a corresponding discussion. As can be seen in the figure, even for this most successful baseline, the results were overall inferior to our approach.  Specifically, SVM-Glove depicted comparable or slightly superior results only for the very small labeled data regime (e.g., when considering 64 labeled instances), and even then, only for 3 out of the 6 topical datasets. Adding more labeled instances, resulted with lower gain for this baseline in all topical datasets. Hence when considering the entire range of fine-tuning sample sizes, its results are inferior to our approach in all 6 topical datasets. Specifically, SVM-Glove results were inferior to BERT-IT-CLUST in 39 out of the 47 data points reported for the 6 topical datasets in Figure 2. In the remaining 4 non-topical datasets, not surprisingly SVM-Glove results were consistently inferior to the BERT-based results, usually by a very large margin.

---

### Decision · Program_Chairs · 2021-01-07
**Final Decision**

**Decision:**

Reject

**Comment:**

The paper suggests a simple variant for BERT training that improves classification for smaller training samples.  So it has a very specific applicability unlike other published variants which generally improve a broad range of tasks.  The variant adds a self-supervision classification task based on clustering.  Experiments are done but it only shows improvement for small training sizes.

AnonReviewer4 suggested a BOW experiment/baseline which was done by the authors in an updated version.  This confirmed the authors line.  AnonReviewer3 asked for computational details, which were added.  AnonReviewer1 lists a number of limitations which the authors need to address and rephrase the statements in their paper.

So it is publishable work, but somewhat marginal due to its specialised nature and thus rejected.